

# Assessing lead and cadmium tolerance of *Chenopodium ambrosioides* during micropropagation: an in-depth qualitative and quantitative analysis

Tour Jan[1], Nasrullah Khan[1], Muhammad Wahab[2], Mohammad K. Okla[3], Mostafa A. Abdel-Maksoud[3], Ibrahim A. Saleh[4], Hashem A. Abu-Harirah[4], Tareq Nayef AlRamadneh[4] and Hamada AbdElgawad[5]

[1] Department of Botany, Faculty of Biological Sciences, University of Malakand, Chakdara, Khyber Pakhtunkhwa, Pakistan
[2] Department of Botany, Faculty of Sciences, Women University Swabi, Swabi, Khyber Pakhtunkhwa, Pakistan
[3] Department of Botany and Microbiology, King Saud University, Riyadh, Saudi Arabia, Saudi Arabia
[4] Department of Medical Laboratory Sciences, Faculty of Science, Zarqa University, Zarqa, Jordan
[5] Integrated Molecular Plant Physiology Research, Department of Biology, University of Antwerp, Antwerp, Belgium

Corresponding author
Nasrullah Khan,
nasrullah@uom.edu.pk

## ABSTRACT

The tolerance of *Chenopodium ambrosioides* to some heavy metals under *in vitro* environment was thoroughly investigated. A micropropagation protocol was developed to facilitate the mass production of plants and to identify metals-tolerant species for potential use in the restoration of polluted areas. Nodal explants exhibited callus formation when treated with $N_6$-benzyladenin (BA) (1.5 mg/l) and a combination of BA/$\alpha$-naphthalene acetic acid (NAA) at concentrations of 1.5/1.0 mg/l on the Murashige and Skoog (MS) medium. The optimal shoot formation was achieved with the callus grown on a medium enriched with 1.5/1.0 mg/l BA/NAA, resulting in an impressive number (21.89) and length (11.79 cm) of shoots. The *in vitro* shoots were rooted using NAA (1.0 and 1.5 mg/l) and were acclimatized in pots with 71% survival rate. After standardizing micropropagation protocol, the *in vitro* shoots were subjected to various doses of lead nitrate (Pb(NO$_3$)$_2$ and cadmium chloride (CdCl$_2$). Pb(NO$_3$)$_2$ and CdCl$_2$ in the media let to a reduction in shoot multiplication, decreasing from 18.73 in the control group to 11.31 for Pb(NO$_3$)$_2$ and 13.89 for CdCl$_2$ containing medium. However, Pb(NO$_3$)$_2$ and CdCl$_2$ promoted shoot length from 5.61 in the control to 9.86 on Pb(NO$_3$)$_2$ and 12.51 on CdCl$_2$ containing medium. In the case of Pb(NO$_3$)$_2$ treated shoots, the growth tolerance index (GTI) ranged from 117.64% to 194.11%, whereas for CdCl$_2$ treated shoots, the GTI ranged from 188.23% to 264.70%. Shoots treated with high level of Pb(NO$_3$)$_2$ induced reddish-purple shoots, while a low level of Pb(NO$_3$)$_2$ induced shoots displayed both green and reddish-purple colors in the same explants. In CdCl$_2$ treated culture, the toxic effects were narrow leaf lamina, elongated petiole and a dark reddish purple coloration. These findings highlight the remarkable potential of *C. ambrosioides* to maintain growth and organogenesis even in the presence Pb(NO$_3$)$_2$ and CdCl$_2$ on the MS medium, indicating a high degree of metal tolerance.

## INTRODUCTION

Soil contamination by heavy metals remains a serious problem and has becoming a great concern due to its adverse effects on humans and ecosystem throughout the world (*Briffa, Sinagra & Blundell, 2020*; *Alengebawy et al., 2021*). In this sense, certain plants have developed the remarkable ability to grow in soil with metal pollutants, accumulating these metals as part of their ecophysiological response in metalliferous environments (*Ahmad & Ashraf, 2011*). Among the hazardous elements, cadmium and lead stand out as notorious heavy metals, exerting adverse effects on plant growth and development due to their toxic nature and widespread existence in the environment (*Jaishankar et al., 2014*; *Jamla et al., 2021*). The impact of these metals extends to the structural, physiological and morphological aspects of plants, leading to growth degradation, wilting and ultimately plant mortality (*Daud et al., 2015*).

Cadmium is a non-degradable contaminant (*Abbas et al., 2017*), swiftly taken up by plants and transported to their above ground part, where it accumulates (*Yang et al., 2010*). This induces multiple symptoms of toxicity in plants, including leaf burning, chlorosis, stunted growth to reduced nutrient absorption (*Kahle, 1993*), disruptions in plant water balance (*Ullah et al., 2020*), and inhibition of stomatal opening (*Bajguz & Hayat, 2009*). Moreover, cadmium can disrupt chlorophyll synthesis, resulting leaf necrosis (*Elazab, Abdel-Wahab & El-Mahdy, 2021*) as well as reduce stem length and dry weight of roots in several plants (*El-Banna & Abdelaal, 2018*). The uptake and absorption of cadmium in plants pose a serious health risk to humans through the food chain (*Shah & Dubey, 1998*). Lead is among the toxic heavy metals (*Zhang, 2003*), occurring naturally in soil but significantly augmented by anthropogenic activities (*Seregin & Ivanov, 2001*).

Plant readily uptake lead from the soil, accumulating it in various plant parts thereby induce stress and other side-effects, such as reduced root elongation and biomass, inhibition of the chlorophyll synthesis, and impairment of certain enzyme activities (*Fargasova, 1994*; *Miranda & Ilangovan, 1996*). The plant's response to lead contamination can vary, depending on the quantity present and the plant genotype exposed to the contaminant. Plant species selectively regarding lead uptake, accumulation ability, and tolerance to lead toxicity may differ (*Janmohammadi, Bihamta & Ghasemzadeh, 2013*; *Topal et al., 2017*). Plants contaminated with heavy metals can enter the animal food chain affecting animal meat and milk quality, and subsequently impacting human food (*Caglarırmak & Hepcimen, 2010*).

*C. ambrosioides* (L.) is an annual shrub belonging to the Chenopodiaceae family, graces the botanical landscape and reaches a maximum height of one meter ($\leq$ 3 ft). This ecologically and medicinally important plant is commonly known as Mexican tea (referred to locally as Skhabotai in northwest Pakistan), and emits a distinctive, albeit unpleasant, aroma. Numerous studies have demonstrated that *C. ambrosioides* is rich in

flavonoids and terpenoids, endowing it with diverse pharmacological properties, including antioxidant and cancer chemopreventive effects (*e.g.*, *Reyes et al., 2019*; *Kiuchi et al., 2002*; *Liu, 2004*). Additionally, it has exhibited increased antioxidant activity in response to Cu-toxicity (*Boojar & Goodarzi, 2007*). Furthermore, an assessment of the *n*-hexane extract of *C. ambrosioides* has revealed molluscicidal activity (*Hmamouchi, Lahlou & Agoumi, 2000*). Nonetheless, the uncontrolled and excessive exploitation of medicinal plants including *C. ambrosioides* for culinary and medicinal purposes in many regions has raised conservation concerns (*Chen et al., 2016*; *Rios et al., 2017*; *Khan et al., 2023*). Likewise, the combined pressures of overexploitation along with the effect of climate change could potentially lead to a decline in the population of *C. ambrosioides*, ultimately threatening its availability for future generations. Thus, it becomes imperative to establish an effective regeneration system to protect the population size of *C. ambrosioides* to ensure its long-term sustainability. The current study presents the first effort to develop an innovative protocol for *in-vitro* propagation of *C. ambrosioides* through callus culture. Additionally, our aim is to assess the plant's response to lead and cadmium stress using *in-vitro* culture as a well-controlled experimental system.

## MATERIALS & METHODS

### Experimental materials

Axillary branches measuring 7–9 cm in length were carefully excised from healthy *C. ambrosioides* plants growing at Malakand University in Khyber Pakhtunkhwa (KP), Pakistan. These branches were subjected to a meticulous cleansing process, involving a through rinsing with tap water for 6–7 min, followed by immersion in a solution of mercuric chloride (0.05%) containing few drops of Tween-20 for 12–15 min. Subsequently, the explants were washed with sterilized distilled water and the surface sterilized explants (2–3 cm long) were then horizontally cultured in screw capped jars containing *Murashige & Skoog (1962)* medium, enriched with 3% sucrose. The medium was semi-solidified with agar (0.6%), and its pH was meticulously adjusted to fall within the range of 5.50–5.56. Following the adjustment, the entire setup was adjusted to autoclaving at a temperature of 121 °C under a pressure of 15 psi for 10–13 min. To provide optimal growth, all cultures were incubated within a growth chamber at a constant temperature of 26 ± 2 °C, under a lighting system that simulated a 16-hr day followed by an 8-hr night cycle.

### Callus induction and subculture for shoot initiation

Explant segments, comprising nodes, leaves, and petioles with sizes ranging from 2 to 3 cm, were placed in culture media containing varying concentrations of NAA (0.5, 1.0, and 1.5 mg/l), Indole-3-acetic acid (IAA) (0.5, 1.0, and 1.5 mg/l), and BA (0.5, 1.0, and 1.5 mg/l). These components were either used individually or combined with NAA (0.5 mg/l) or IAA (0.5 mg/l) to initiate the formation of callus (Table 1). After a thirty-day period, calluses that had developed from nodes, leaves, and petioles were isolated and subsequently transferred to growth media containing different concentrations of NAA (1.0, 1.5, and 2.0 mg/l), BA (1.5 and 2.0 mg/l), and combinations of BA/NAA (1.5/1.0 and

**Table 1 Callus initiation from node, leaf and petiole segment.**

| PGR | Conc (mg/l) | Node %Re. | Node AC (mg) | Node Characteristic | Leaf %Re. | Leaf AC (mg) | Leaf Characteristics | Petiole %Re. | Petiole AC (mg) | Petiole Characteristics |
|---|---|---|---|---|---|---|---|---|---|---|
| NAA | 0.5 | 57.46[b] | 75 | Reddish purple, brownish, compact, granular | 62.46[b] | 82 | Reddish purple, granular, soft | 58.46[c] | 102 | Creamy, Brownish, soft, granular |
|  | 1.0 | 63.82[a] | 138 |  | 57.82[c] | 163 |  | 73.82[a] | 156 |  |
|  | 1.5 | 65.27[a] | 127 |  | 76.37[a] | 151 |  | 78.27[a] | 135 |  |
| IAA | 0.5 | 47.32[b] | 67 | Light Reddish purple, soft, granular | 54.73[c] | 69 | Reddish purple, soft, Brownish, granular | 51.21[c] | 97 | Dark brownish, soft, granular |
|  | 1.0 | 52.93[b] | 122 |  | 64.78[b] | 138 |  | 68.32[b] | 175 |  |
|  | 1.5 | 64.02[a] | 131 |  | 68.21[b] | 152 |  | 64.72[b] | 179 |  |
| BA | 0.5 | 40.23[c] | < | Light yellowish compact, granular, light, hard | – | – | – | – | – | – |
|  | 1.0 | 45.18[b] | < |  | – | – |  | – | – |  |
|  | 1.5 | 51.72[c] | < |  | – | – |  | – | – |  |
| BA/ NAA | 0.5/0.5 | 57.38[c] | 102 | Light yellowish, friable, granular | 66.35[b] | 113 | Creamy, Reddish purple, granular, soft | 53.28[c] | 121 | Creamy, brownish, soft and friable |
|  | 1.0/0.5 | 64.71[a] | 162 |  | 58.32[c] | 159 |  | 65.16[b] | 189 |  |
|  | 1.5/0.5 | 59.85[b] | 157 |  | 58.72[c] | 192 |  | 73.38[a] | 198 |  |
| BA/IAA | 0.5/0.5 | 49.86[b] | 86 | Reddish purple, soft, granular | 47.83[d] | 95 | Light Reddish purple, soft, granular | 58.29[c] | 87 | Light brownish, soft, granular |
|  | 1.0/0.5 | 72.31[a] | 158 |  | 67.46[b] | 179 |  | 66.97[b] | 167 |  |
|  | 1.5/0.5 | 60.39[a] | 149 |  | 62.91[b] | 173 |  | 73.31[a] | 183 |  |
| Control | 0.0/0.0 | – | – | – | – | – | – | – | – | – |

Notes.

PGR, Plant growth regulators; Re, Response; AC, Amount of callus; <, less than 55; –, Zero.

Mean followed by different letters are statistically significant at $p<0.05$.

2.0/1.0 mg/l). These conditions were utilized to induce the formation of shoots through sub-culturing.

## Shoot proliferation and root induction

For shoot proliferation, the *in-vitro* shoots were sliced into two to three pieces and transferred onto a growth medium supplemented with varying concentrations of growth regulators. Specifically, concentrations of 1.0, 1.5, and 2.0 mg/l of BA and kinetin were used. These growth regulators were applied either individually or in combination with NAA at a concentration of 1.0 mg/l. The aim was to standardize the concentrations and combinations for shoot proliferation (Table 2). The observation took place four weeks after the initial inoculation, where the shoot multiplication percentage, as well as the number and length of shoots, were recorded.

Moving on to root induction, individual shoots that were 30 days old were carefully excised and transferred to a medium containing a combination of BA and NAA at concentrations of 1.5/1.0 mg/l for the purpose of regeneration. Subsequently, these shoots were further placed on media supplemented with NAA at concentrations of 1.0, 1.5, and 2.0 mg/l. This was done both individually and in combination with BA/NAA at 1.5/1.0 mg/l and kinetin/NAA at 1.5/1.0 mg/l. The intention here was to facilitate the induction of roots. The progress of root formation, including induction percentage, numbers, and length, were then observed and recorded on the 24th and 26th days post-inoculation.

**Table 2  Shoots initiation from callus at different concentrations/combinations of PGR.**

| PGR | Conc. (mg/l) | Node derived Callus | | Leaf derived Callus | | Petiole derived callus | |
|-----|---|-----|-----|-----|-----|-----|-----|
| | | NS ± Sd | LS (cm) ± Sd | NS | LS | NS | LS |
| NAA | 1.0 | 6.25 ± 0.61[c] | 3.21 ± 0.39[a] | NR | NR | NR | NR |
| | 1.5 | 6.39 ± 0.34[c] | 4.04 ± 0.78[a] | ,, | ,, | ,, | ,, |
| | 2.0 | 5.45 ± 1.73[c] | 2.56 ± 0.26[a] | ,, | ,, | ,, | ,, |
| BA | 1.5 | 10.84 ± 1.45[a] | 5.21 ± 0.83[b] | ,, | ,, | ,, | ,, |
| | 2.0 | 14.83 ± 0.95[b] | 5.42 ± 0.08[b] | ,, | ,, | ,, | ,, |
| BA/NAA | 1.5/1.0 | 9.43 ± 0.63[a] | 3.73 ± 0.29[a] | ,, | ,, | ,, | ,, |
| | 2.0/1.0 | 8.78 ± 0.49[a] | 3.34 ± 0.84[a] | ,, | ,, | ,, | ,, |

**Notes.**
NS, Number of shoots; LS, Length of shoots; NR, Not responding; ± Sd, Standard deviation.
Mean followed by different alphabets are statistically significant at $p < 0.05$.

## Acclimatization

Plantlets were gradually acclimated to a non-sterile environment by gently loosening the screws of the culture containers and lastly detached the cap. The plantlets were detached from the culture media and rinsed using sterilized distilled water, effectively eliminating the residual observing medium. These cleansed plantlets were then transplanted into plastic cups filled with a sterilized mixture of soil and sand (2:1 ratio). To initiate this transition, the plantlets were initially enclosed with plastic cups and wetted with half MS medium.

## Heavy metals treatments and growth tolerance index (GTI)

In the context of heavy metals treatments, the *in-vitro* shoots were transplanted onto a growth medium enriched with specific metals, in conjunction with the optimal blend of BA/NA at a ratio of 1.5/1.0 mg/l. A series of concentrations for $Pb(NO_3)_2$ (0.5, 1.5, and 2.5 mg/l) and $CdCl_2$ (0.5, 1.5, and 2.5 mg/l) were employed, combined with the previously mentioned optimal BA/NAA ratio (1.5/1.0 mg/l). The purpose of this study was to identify strains that exhibited a degree of tolerance. As a control group, an MS medium was employed that contained the ideal BA/NAA ratio (1.5/1.0 mg/l), yet lacked the introduction of $Pb(NO_3)_2$ and $CdCl_2$. Subsequent to these treatments, the *in-vitro* shoots underwent assessment in terms of measurements and weight determination. To ascertain the dry biomass, the plant materials underwent a 24-hour exposure to 80 °C and were subsequently weighed. This facilitated the computation of the growth tolerance index (GTI) for the shoots, presented as a percentage. The GTI was calculated using the following formula:

$$GTI = \frac{\text{Dry weight of shoots developed on media with } Pb(NO_3)_2 \text{ or } (CdCl_2)}{\text{Dry weight of shoots developed on media devoid of } Pb(NO_3)_2 \text{ or } (CdCl_2)} \times 100.$$

## Statistical analysis

Differences among the treatments were assessed using a single factor analysis of variance (ANOVA). Furthermore, the variances among the variables were examined by a *post-hoc* Tukey honest significant difference (HSD) test. Statistically significant variations were

considered significant at a threshold of $P < 0.05$. To assess the impact of heavy metals, we conducted a correlation analysis involving multiple parameters of *C. ambrosioides* such as the concentration of heavy metals (HMC mg/l), the growth tolerance index (GTI %), multiplication rate (%), number of shoots, shoot length, and biomass. All the statistical analyses were performed in XLstat and OriginLab Pro-2023b (https://www.originlab.com) software's.

## RESULTS

### Callus induction

Node, leaf and petiole of *C. ambrosioides* were inoculated on media containing various concentrations of NAA, IAA, BA, NAA/BA, IAA/BA and NAA/IAA as outlined in Table 1. Within a span of 10 days from inoculation, all three types of explants exhibited the development of callus. The texture of resulting callus varied, ranging from compact and soft to friable and granular. Specifically, callus formation occurred in petiole explant at a rate of 78.27%, leaf at a rate of 76.37% when treated with 1.5 mg/l NAA, and in node at a rate of 65.71% with 1.0/0.5 mg/l BA/NAA (Table 1). Notably, the callus developed from nodes, particularly under the influence of BA with or without presence of NAA, exhibited a remarkable organogenic potential. Conversely, the callus derived from petiole and leaf explants did not display any organogenic potential (Table 2). In the case of IAA, both alone and in combination with BA, the formation of callus was observed; however, their corresponding fresh weights and percentage responses remained relatively low (Table 1). Explants that were placed on a growth regulator-free medium (control) failed to exhibit any growth and eventually died after 15 days of inoculation.

### Shoot from callus

Thirty days old callus from node, leaf and petiole were isolated and sub-cultured on media containing NAA, BA and BA/NAA (Table 2). Nodal explants produced compact, friable, glandular callus with a light yellowish type. In the presence of BA, either alone or with conjunction with NAA (Table 2), these nodal callus cultures displayed prolific shoot formation. The average number of shoot formed with BA ranged from $10.84 \pm 1.45$ to $14.83 \pm 0.93$, with corresponding shoot lengths ranging from $5.21 \pm 0.83$ to $5.42 \pm 0.83$. Conversely, when utilizing the combination of BA/NAA, the average number of shoots formed ranged from $9.43 \pm 0.63$ to $8.78 \pm 0.49$. Notably, the application of NAA (at concentration of 1.0–2.0 mg/l) resulted in the formation of an average of $6.25 \pm 0.61$ to $5.45 \pm 1.73$ shoots, respectively. Nevertheless, callus derived from leaf and petiole sources exhibited a lack of shoots regeneration (Table 2).

### Multiplication of shoots

Cluster of shoots (2–3) were sub-cultured on media having various concentrations and combinations of BA, Kn or BA/NAA and Kn/NAA in order to standardized the optimal concentration and combination for shoot multiplication (Table 3). The results of shoots multiplication were recorded for all concentrations of BA and Kn, both with or without NAA (Table 3). Notably, the application of BA/NAA induced the highest number of shoots,

**Table 3  Effect of various concentration/combinations of PGR on shoots multiplication of *C. ambrosioides*.**

| PGR | Concentration (mg/l) | Number of shoots ± Sd | Length of shoots (cm) ± Sd |
|---|---|---|---|
| | 1.0/1.0 | 18.73 ± 1.67[a] | 5.26 ± 0.27[a] |
| BA/NAA | 1.5/1.0 | 21.89 ± 1.94[a] | 5.49 ± 0.84[a] |
| | 2.0/1.0 | 18.39 ± 1.68[a] | 4.82 ± 0.76[a] |
| | 1.0 | 16.57 ± 1.03[b] | 3.65 ± 0.72[b] |
| BA | 1.5 | 16.49 ± 0.82[b] | 4.05 ± 0.32[b] |
| | 2.0 | 14.61 ± 0.61[b] | 3.95 ± 0.84[b] |
| | 1.0/1.0 | 11.43 ± 0.72[c] | 3.13 ± 0.39[c] |
| Kn/NAA | 1.5/1.0 | 9.87 ± 0.54[c] | 3.47 ± 0.48[c] |
| | 2.0/1.0 | 9.35 ± 0.39[c] | 2.78 ± 0.07[c] |
| | 1.0 | 7.83 ± 0.86[c] | 2.58 ± 0.09[c] |
| Kn | 1.5 | 7.95 ± 0.35[c] | 2.47 ± 0.41[c] |
| | 2.0 | 8.46 ± 0.08[c] | 1.85 ± 0.15[c] |
| Control | – | – | – |

Notes.
[a,b,c] Mean followed by different letters are statistically significant at $p < 0.05$.

**Table 4  Rooting of *in vitro* shoots in the influence PGR.**

| PGR | Concentration (mg/l) | % Rooted shoot | Roots number/ explant ± Sd | Roots length (cm) ± Sd |
|---|---|---|---|---|
| | 1.0 | 76.73[a] | 12.32 ± 0.58[a] | 9.34 ± 0.42[b] |
| NAA | 1.5 | 68.02[b] | 11.61 ± 0.47[a] | 12.81 ± 0.72[a] |
| | 2.0 | 64.89[b] | 9.89 ± 0.09[b] | 10.79 ± 0.91[b] |
| BA/NAA | 1.5/1.0 | 40.21[c] | 8.07 ± 0.16[b] | 13.23 ± 1.03[a] |
| Kn/NAA | 1.5/1.0 | – | – | – |

Notes.
[a,b,c] Mean followed by different alphabets are statistically significant at $p < 0.05$.

ranging from 21.89 ± 1.94 to 18.39 ± 1.68, with their shoot lengths ranged from 5.49 ± 0.84 to 4.82 ± 0.76 cm. The supplementation of BA alone proliferated shoots in the ranged of 16.5 ± 1.03 to 14.61 ± 0.61. Interestingly, Kn either alone or in combination, produced less number of shoots compared to BA, whether applied separately or in combination (Table 3).

### Root induction

The best rooting rate (76.73%) and number (12.32 ± 0.58) were recorded in the medium comprising NAA at a concentration of 1.0 mg/l. The formation of roots with BA/NAA (1.5/1.0 mg/l) resulted in a lower number (8.07 ± 0.16) but higher length (13.23 cm). However, the medium supplemented with Kn/NAA did not support roots formation (Table 4).

### Acclimatization

The survival potential of the *in-vitro* regenerated plantlets when transferred to the *ex-vitro* environment was 71% (Fig. 1). Notably, there were no obvious phenotypic differences

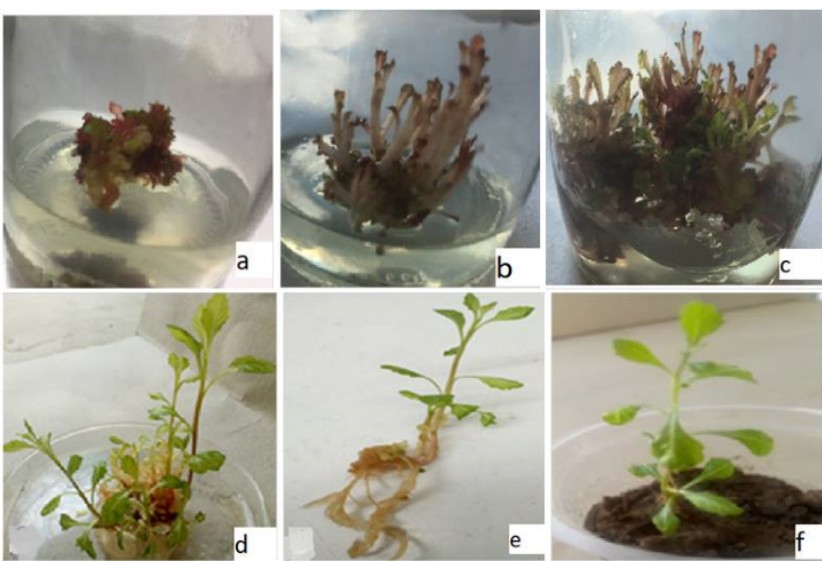

**Figure 1** *In-vitro* **propagation of C. ambrosioides.** (A) Shoots from callus, (B) abnormal shoots with BA/CdCl2, (C) abnormal and normal shoots with BA/Pb(NO3)2, (D) normal shoots with BA/NAA, (E) roots with NAA, (F) acclimatized plant.

observed between the acclimatized plants and the original mother stock. Subsequently, the two month old plantlets were then transplanted into the lath house.

## Heavy metals and shoot proliferation

After optimizing of culture condition, subsequent experiments were designed to identify a suitable line for metals ($Pb(NO_3)_2$ and $CdCl_2$) tolerance (Table 5). The inclusion of $Pb(NO_3)_2$ in the medium inhibited shoots multiplication rate. The percentage of shoot multiplication per explants dropped from 86.74% in the control treatment to 57.38% in culture treated with $Pb(NO_3)_2$ (Table 5). Although, $Pb(NO_3)_2$ containing media did not exhibit any visible symptoms of phenotypic alteration except for a reddish purple color, all shoots maintained a normal morphology characterized by broad lamina and short petiole similar to those of the mother plant. In some instances, the same explants produced both normally morphed shoots with green and reddish purple color shoots (Fig. 1C) at 0.5 mg/l of $Pb(NO_3)_2$ in the medium. The growth of the shoot biomass (dry weight) remained unaffected by $Pb(NO_3)_2$ at 0.5 mg/l in the medium and was comparable to the control, while higher concentration $Pb(NO_3)_2$ in the medium led to an increase biomass growth (Table 5).

The culture of *C. ambrosioides* vigorously reproduced *in-vitro* shoots proliferation in media supplemented with $CdCl_2$. The highest propagation percentage (75.71%) was obtained with 0.5 mg/l of $CdCl_2$ concentration in the medium. The shoots formed under varying concentrations of $CdCl_2$ ranged from $16.31 \pm 1.61$ to $13.89 \pm 1.94$ per explants. Notably, in presence of lower $CdCl_2$, concentrations, shoot multiplication was as high as in the control (Table 5). Shoots with 0.5 mg/l $CdCl_2$ displayed a normal morphology, while

Table 5 Effect of various doses of metals on multiplication and growth of shoot in the existence of BA/NAA (1.5/1.0 mg/l).

| Metals (mg/l) Lead | Cadm. | Multiplication % | Number of shoots ± Sd | Length of shoot ± Sd | Toxicity symptoms | Shoots dry weight (mg) ± Sd |
|---|---|---|---|---|---|---|
| 0.5 | | 57.38 | 14.57 ± 2.03[a] | 5.16 ± 1.29[c] | In few illustrations green and reddish purple branches with normal leaf and petiole were noted. | 20 ± 1.8[a] |
| 1.5 | | 68.71 | 11.69 ± 2.82[b] | 7.52 ± 1.44[b] | Shoots were reddish purple color with broad leaf lamina and short petiole as in the mother plants. | 26 ± 2.3[b] |
| 2.5 | | 59.85 | 11.31 ± 1.61[b] | 9.86 ± 2.36[a] | | 33 ± 2.9[b] |
| | 0.5 | 75.71 | 16.31 ± 1.61[a] | 6.36 ± 1.47[c] | Normal | 32 ± 2.08[b] |
| | 1.5 | 63.38 | 13.89 ± 1.94[b] | 8.15 ± 1.39[b] | Reddish purple color shoots with disturbed morphology (narrow leaf lamina and elongated petiole) | 37 ± 2.6[b] |
| | 2.5 | 59.25 | 14.09 ± 1.68[b] | 12.51 ± 2.73[a] | | 41 ± 3.35[b] |
| Control | | 86.74 | 18.73 ± 1.67[a] | 5.61 ± 1.46[c] | Normal | 17 ± 2.0[a] |

Notes.
[a,b,c]Mean followed by different alphabets are statistically significant at $p < 0.05$.

Table 6 GTI for the *in-vitro* shoots of *C. ambrosioides* in the presence metals.

| Treatment Lead | Cadmium | (GTI)(%) |
|---|---|---|
| 0.5 | | 117.64 |
| 1.5 | | 152.94 |
| 2.5 | | 194.11 |
| | 0.5 | 188.23 |
| | 1.5 | 217.64 |
| | 2.5 | 264.70 |

1.5 and 2.5 mg/l of CdCl$_2$ induced morphological abnormalities in the *in-vitro* shoots. The abnormalities included narrow leaf lamina, elongated petioles, and reddish purple coloration (Fig. 1B). Biomass growth of shoots exhibited an increased trend with increasing concentration of CdCl$_2$ (Table 5).

## Heavy metals and growth tolerance

The growth tolerance index (GTI) calculation revealed that shoot growth was enhanced in the presence of heavy metals (Table 6). The highest GTI value was observed in the medium comprising 2.5 mg/l of lead, reaching to 194.11%. Conversely, lower doses of lead resulted into low dry biomass production, with GTI values of 152.94% for 1.5 mg/l and 117.64% for 0.5 mg/l. In contrast, the GTI exhibited a reduction in culture grown on cadmium-containing medium compared to lead-containing medium. The GTI values for cadmium-containing media ranged from 188.23% to 264.70% as given in Table 6. Furthermore, the heavy metals concentration has a significant positive ($r = 0.671$; $p < 0.01$) effect on GTI (%), shoot length ($r = 0.924$; $p < 0.001$) and dry weight (($r = 0.651$; $p < 0.01$) respectively (Fig. 2). In contrast, the HMC (mg/l) presence in the media exhibited an adverse effect on multiplication ($r = -0.24$; $p < 0.05$) and number of shoots ($r = -0.655$; $p < 0.01$).

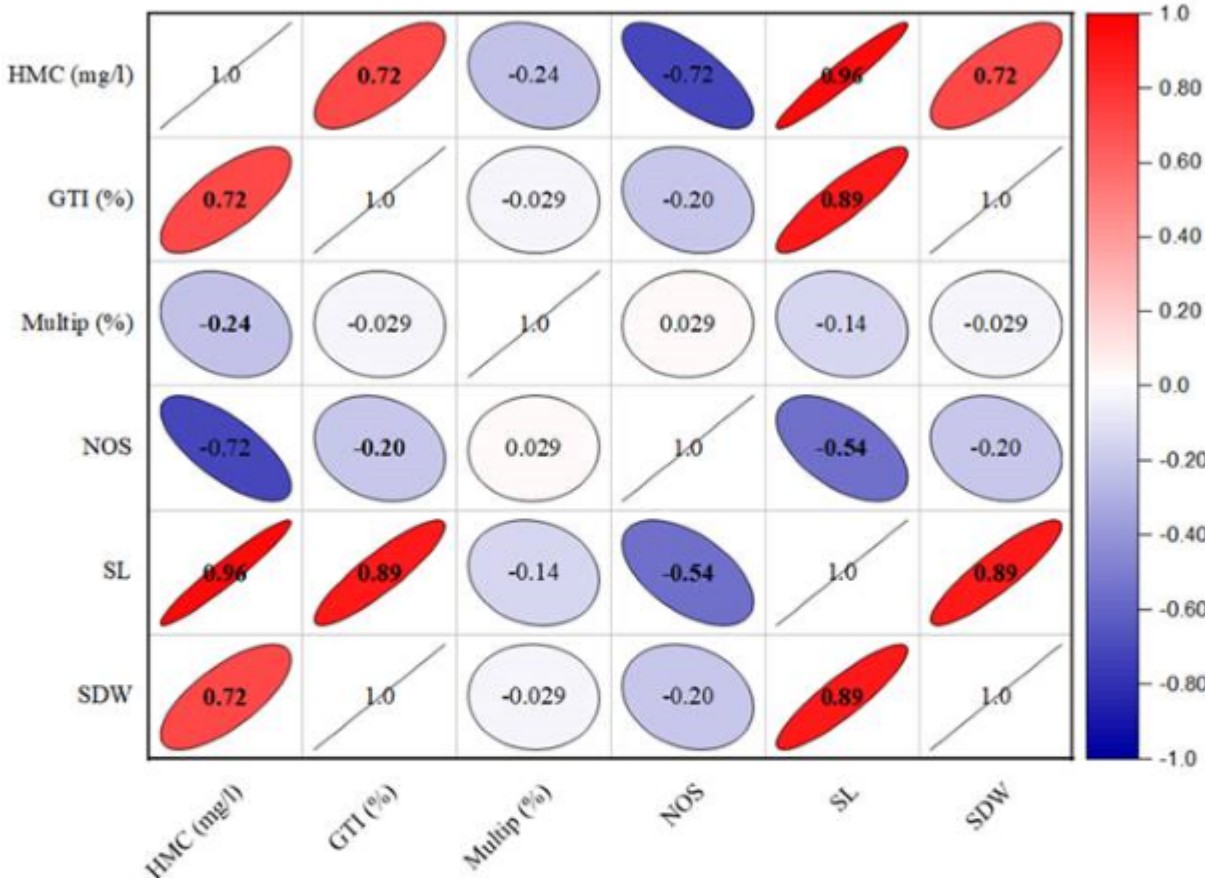

**Figure 2** The effect of heavy metals concentration on various structural attributes of *C. ambrosioides*.

## DISCUSSION

Plant tissue culture provides several advantages for studying the metabolic capabilities of plant cells and their capacity for toxicity tolerance. Understanding the impacts of plant cells on pollutant uptake and purification, free from microbial interference, hold significant importance in the pursuit of essential knowledge about plants (*Doran, 2009*). This study aimed to establish an effective *in vitro* protocol for the large-scale multiplication of *C. ambrosioides* through callus culture. Additionally, it sought to assess the tolerance ability of plant material to lead and cadmium stress using aseptic *in-vitro* culture techniques. In our studies, the morphogenic responses of the three types of explant showed variation based on the concentration and combinations of growth regulators employed. Nodal explants exhibited the development of multiple shoots within the callus, while leaf and petiole explants proved conducive to callus formation. Prior research, such as the study conducted by *Hesami & Daneshvar (2016)* on C. quinoa, as well as *Nathar & Yatoo (2014)* on *Artemisia pallens*, highlighted the superior suitability of hypocotyle for *C. quinoa* and shoot tip, leaf, and petiole for *Artemisia pallens* as explants for callus formation. Our study contributes by documenting the formation of compact, friable, granular callus with a light
yellowish type from node, leaf and petiole explants under the influence of either BA alone or in combination with NAA. It is north noting, that unlike the nodal explant, the callus derived from leaf and petiole has not been able to demonstrate organogenic potential.

The difference in shoots formation between different types of explants in response to BA could be linked to the level of endogenous cytokinins as suggested by *Yucesan, Turke & Gure (2007)*. Our results exposed that, in the case *C. ambrosioides*, BA exhibited its cytokinin activity most effectively at lower concentration, facilitating multiplication of new shoots. However, as the concentration of BA increased, the numbers of newly formed shoots show a decrease. This observation aligns with the study of *Marino et al. (1993)* which highlighted a similar trend in apricot shoots growth. Specifically, they noted an increased in shoot numbers with increasing BA levels up to 2.0 mg/l, beyond which a subsequent rise in concentration led to a reduction shoot multiplication. For achieving optimal shoots multiplication in our study, a moderate concentration of cytokinin (BA) at 1.5 mg/l and a low level of auxin (NAA) at 1.0 mg/l were found to be necessary.

Morphologically, the shoots that developed under this specific combination exhibited normal characteristics including green shoots, well-formed leaves and healthy petioles. The significance of utilizing a combination of growth regulators has been emphasized in several studies. *Raihana et al. (2011)* produced the maximum number of shoot proliferation in *Curcuma manga* through the application of BA and NAA, while *Bharalee, Das & Kalita (2005)* achieved similar results in *Curcuma caesia*. In our root induction experiments, the *in-vitro* shoots of *C. ambrosioites* also displayed root formation on the same medium (BA/NAA) used for shoot multiplication. Additionally, roots were successfully induced using media containing varying concentration of NAA. These results are consistent with the outcomes reported by *Mongkolsawat et al. (2018)* and *Jan et al. (2020)*.

Several researchers aspire to provide essential nutritious food and fodder for the future (*White, 2012*). But ensuring the safety of both food and fodder is of paramount importance. The toxic properties of Cd have been extensively studied and ongoing research is now centered on the cultivation of crops that can withstand Cd presence in the environment. Additionally, efforts have been made to cultivate plants with the capacity to accumulate Cd for purpose phytoremediation in polluted lands (*White, 2012*; *Martinka, Vaculík & Lux, 2014*; *White & Pongrac, 2017*). The achievement of successful phytoextraction is commonly linked to a plant's capability to endure the presence of of pollutant in the substrate (*Di-Lonardo et al., 2011*).

To determine the tolerance ability, the *in-vitro* shoots of *C. ambrosioides* were exposed to different concentrations of lead and cadmium. In the eco-toxicological evaluation, it is essential to avoid lethal doses of the metals toxicity in order to select the tolerant line, its concentration has to be appropriately raised (*Wierzbicka et al., 2007*; *Doran, 2009*). Considering the observed tolerance of the evaluated *C. ambrosioides*, it can be concluded that there was no shoot necrosis on any media containing lead or cadmium. This absence of necrosis suggests the evaluated concentrations of metals did not indicate pollution which has also been advocated by *Katanic et al. (2007)*. In our experiments, the application of lead treatment resulted in a notable suppression of the proliferation percentage, leading to a reduction of 57.38% at 0.5 mg/l concentration. Conversely, at the same concentration

(0.5 mg/l), cadmium exposure triggered a proliferation rate of 75.71%. Interestingly *Alyssum montanum* exhibited enhanced multiplication, rooting, and biomass production at low to moderate levels of cadmium, as observed by *Wiszniewska et al. (2017)*.

The encouraging impact of lead has also been reported by *Seregin & Ivanov (2001)* on root multiplication. The increase in shoot length was observed across all concentrations of Pb and Cd in the growth media; however the positive effect of Cd was more pronounced compared to Pb. In our research, the increase in shoot length and dry biomass served as useful indicators for assessing metal tolerance in plants. The phenomenon of metals working synergistically to promote plant growth was similarly reported by *John et al. (2008)*. *Kalisov-Spirochova et al. (2003)* also noted a positive effect of Pb on total biomass accumulation in aspen rooted-shoots cultured *in vitro*. Nevertheless, *Katanic et al. (2007)* reported an adverse effect on the fresh biomass of white poplar shoot tips when grown on the multiplication medium enriched with Pb.

In our study, Pb and Cd induced phenotypic alteration in shoots at higher doses. Highest doses of Pb induced reddish purple color shoot while low dose induced green and reddish purple shoots from the same explant. It was observed, that high Cd stress induced greater toxicity and the visible toxic symptoms were narrow leaf lamina and short petiole and reddish purple coloration. Similar toxicity symptom has been reported by *Pandey & Sharma (2002)* in cabbage plant. The stunted growth rate in plants at high dose of cadmium may be because of a disorder in the enzymatic activities, transpiration and photosynthesis (*Almeida et al., 2007*). *C. ambriosoides* maintained biomass production on Pb and Cd selection system of shoots what can be considered an indicator of metals tolerance (*Gomes, Marques & Soares, 2013*). Comparing the tolerance index, *C. ambriosoides* tolerant to Cd concentrations compared to Pb. Our findings are accordance with the findings of other studies (*e.g.*, *Zacchini et al., 2009*; *Dos-Santos et al., 2007*).

## CONCLUSIONS

Plant tissue culture techniques, such as *in vitro* shoot cultures, play central role in phytoremediation studies. In the case of the studied *C. ambrosioides*, it exhibited a greater capacity to maintain growth and organogenesis in medium containing $CdCl_2$ compared to $Pb(NO_3)_2$. The influence of cadmium on the plant's development surpassed that of lead, resulting in enhanced shoot length, greater shoot dry weight, and notable phenotypic alterations within the *in-vitro* shoot structure. Thus, *C. ambrosioides* display a higher tolerance towards $CdCl_2$ than $Pb(NO_3)_2$, as evidenced by its elongated shoots, total plantlet biomass, and GTI and may be used as a candidate plant for phytoremediation.

## ACKNOWLEDGEMENTS

The authors gratefully acknowledge the Department of Botany, University of Malakand Chakdara Dir (L) Khyber Pakhtunkhwa, Pakistan for generously providing the laboratory facilities that were essential for the successful completion of this work.

### Funding

The authors received no funding for this work.

### Competing Interests

The authors declare there are no competing interests.

### Author Contributions

- Tour Jan conceived and designed the experiments, performed the experiments, prepared figures and/or tables, authored or reviewed drafts of the article, and approved the final draft.
- Nasrullah Khan conceived and designed the experiments, performed the experiments, prepared figures and/or tables, authored or reviewed drafts of the article, and approved the final draft.
- Muhammad Wahab conceived and designed the experiments, authored or reviewed drafts of the article, and approved the final draft.
- Mohammad K. Okla analyzed the data, authored or reviewed drafts of the article, and approved the final draft.
- Mostafa A. Abdel-Maksoud analyzed the data, authored or reviewed drafts of the article, and approved the final draft.
- Ibrahim A. Saleh analyzed the data, authored or reviewed drafts of the article, and approved the final draft.
- Hashem A. Abu-Harirah conceived and designed the experiments, prepared figures and/or tables, and approved the final draft.
- Tareq Nayef AlRamadneh analyzed the data, prepared figures and/or tables, and approved the final draft.
- Hamada AbdElgawad conceived and designed the experiments, prepared figures and/or tables, and approved the final draft.

### Data Availability

The raw data are available in the Supplemental Files.

### Supplemental Information

Supplemental information for this article can be found online at http://dx.doi.org/10.7717/peerj.16369#supplemental-information.

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
