# Peer review of "Assessing lead and cadmium tolerance of Chenopodium ambrosioides during micropropagation: an in-depth qualitative and quantitative analysis"

_PeerJ, doi:10.7717/peerj.16369_

## Round 0.1 · original submission · Major Revisions

Dear Authors,

I request you to revise the manuscript in accordance with the suggestions provided by the reviewers.

Reviewer 1 ·

Basic reporting

The manuscript “Tolerance to lead and cadmium of Chenopodium ambrosioides during micropropagation” deals with an interesting investigation has been conducted into the tolerance of Chenopodium ambrosioides to heavy metals under in vitro conditions. In order to remedy polluted areas in the future, the use of micropropagation protocols for mass propagation is vital for subsequent applications in remediating polluted sites. In this study, the authors successfully standardized the micropropagation protocol using nodal explants, resulting in a successful callus formation and a successful shoot multiplication in the plants. The combination of BA and NAA in the medium has proved to be effective in optimizing shoot number and length, which is a significant finding when it comes to mass propagation of plants. A promising step towards the possibility of applying this technology to the field is the acclimatization of in vitro shoots into pots with a 71% survival rate. The detailed comments are provided below:

Comments
• The introduction section is very short. Kindly elaborate the content with more facts support the present research work. It would be valuable to investigate the translocation and accumulation of heavy metals within the plant tissues to assess potential phytoremediation capabilities in the introduction section.
• Provide the latest reference. Presently, the reference is very old.
• LN 46-51: Please rewrite the line.
• LN 104: Which software is used for analysis should be mentioned in the statistical analysis section.
• The result should be discussed with the relevance of the statistical analysis.
• The authors should consider providing more detailed information on the methods used to quantify heavy metal concentrations in the plant tissues. Specifically, information on the analytical techniques, such as atomic absorption spectroscopy would strengthen the credibility of the results and help readers understand the accuracy of the metal concentration measurements.
• In the discussion section the potential application of the micropropagation protocol for other plant species with phytoremediation potential could be discussed.
• The growth tolerance index (GTI) values ranging from 117.64% to 194.11% in Pb(NO3)2 treated shoots and 188.23% to 264.70% in CdCl2 treated shoots indicate a significant degree of variability in the plant's response to heavy metal exposure. It would be insightful to analyze the correlation between GTI values and the specific concentrations of heavy metals in the medium to determine the threshold levels at which the plant's growth is significantly affected. This information could aid in selecting the most tolerant genotypes for phytoremediation efforts.
• A discussion on the potential implications of heavy metal tolerance in Chenopodium ambrosioides for food chain interactions and wildlife should be included.
• Addressing the possible effects of heavy metal exposure on the nutritional quality of the plant would be relevant, especially if used for phytoremediation and subsequent consumption.
• Conclusion section is very poorly discussed.
• A concise and informative conclusion should summarize the main findings and their significance in the context of the broader scientific community and environmental applications.

Experimental design

Please see the comment above

Validity of the findings

Please see the comment above

Additional comments

Please see the comment above

Reviewer 2 ·

Basic reporting

The title of the study does not adequately describe the contents. To accurately reflect the study, please provide a clear and concise title.
Introduction is too short. What is the current knowledge about Chenopodium ambrosioides and heavy metal tolerance please elaborate in introduction?
Line 58: Axillary branches (7-9 cm long) were excised from healthy plant? Mention name of plant and source.
Study objectives are not clearly defined. Are you primarily focused on the development of a micropropagation protocol for Chenopodium ambrosioides, or are you investigating its potential as a metal-tolerant plant?

Experimental design

Several sections of material and methods may be either merged or properly elaborated.
More details about the experimental statistical analysis should be provided. The results section does not include any information about statistical analysis.

Validity of the findings

When using abbreviations, make sure to define them at their first use.
There seems to be a lack of a clear discussion section, where the implications of the findings are interpreted and placed in context of the existing literature.
In conclusion, the paper should summarize the main findings, their implications, and potential future research directions. Can Chenopodium ambrosioides act as a phytoremediator?

---

## Round 0.2 · Minor Revisions

The manuscript is revised substantially but requires minor modifications identified by the Section Editor:

- Title typo: "Quantitative and Quantitative" (one of these should be "Qualitative"??

- line 16: cadmium and lead should not be capitalized ++ line 70-71: Grammatically incorrect

- line 71-71: "known locally as..." local to where? Also isn't this plant native to Central and South America, not Pakistan?

- line 73-74: The reference to Sa, 2016 is incorrect as Sa is not the primary source for this information; Sa cites a different source. In addition, the Sa source is misquoted. Sa says "one of the most used" not "the most widely used"

- line 77 "Therefore...". This is a non sequitur. Is the species threatened? And if it is, is in vitro propogation the best way to protect it?

Reviewer 1 ·

Basic reporting

The authors made significant changes in the manuscript as per the suggestion of the reviewer.

Experimental design

See section 1

Validity of the findings

See section 1

Additional comments

See section 1

Reviewer 2 ·

Basic reporting

The manuscript has been revised nicely and it can be accepted in its current state.

Experimental design

NA

Validity of the findings

NA

Additional comments

NA

---

## Round 0.3 · accepted · Accept

The authors have revised the manuscript as suggested. The manuscript can be accepted in its current form.